# BLIND PARETO FAIRNESS AND
# SUBGROUP ROBUSTNESS

## ABSTRACT

With the wide adoption of machine learning algorithms across various application domains, there is a growing interest in the fairness properties of such algorithms. The vast majority of the activity in the field of group fairness addresses disparities between *predefined* groups based on protected features such as gender, age, and race, which need to be available at train, and often also at test, time. These approaches are static and retrospective, since algorithms designed to protect groups identified *a priori* cannot anticipate and protect the needs of different at-risk groups in the future. In this work we analyze the space of solutions for worst-case fairness beyond demographics, and propose *Blind Pareto Fairness* (BPF), a method that leverages no-regret dynamics to recover a fair minimax classifier that reduces worst-case risk of *any* potential subgroup of sufficient size, and guarantees that the remaining population receives the best possible level of service. BPF addresses *fairness beyond demographics*, that is, it does not rely on predefined notions of at-risk groups, neither at train nor at test time. Our experimental results show that the proposed framework improves worst-case risk in multiple standard datasets, while simultaneously providing better levels of service for the remaining population, in comparison to competing methods.

## 1 INTRODUCTION

A large body of literature has shown that machine learning (ML) algorithms trained to maximize average performance on existing datasets may present discriminatory behaviour across pre-defined demographic groups (Barocas & Selbst (2016); Hajian et al. (2016)), meaning that segments of the overall population (data) are measurably under-served by the ML model. This has sparked interest in the study on why these disparities arise, and on how they can be addressed (Mitchell et al. (2018); Chouldechova & Roth (2018); Barocas et al. (2019)). One popular notion is *group fairness*, where the algorithm has access to a set of predefined demographic groups during training, and the goal is to learn a model that satisfies a certain notion of fairness across these groups (e.g., statistical parity, equality of opportunity) (Dwork et al. (2012); Hardt et al. (2016)); this is usually achieved by adding a constraint to the standard optimization objective. It has been shown that optimality may be in conflict with some notions of fairness (e.g., the optimal risk is different across groups) (Kaplow & Shavell (1999); Chen et al. (2018)), and perfect fairness can, in general, only be achieved by degrading the performance on the benefited groups without improving the disadvantaged ones. This conflicts with notions of no-harm fairness such as in (Ustun et al. (2019)), which are appropriate where quality of service is paramount. Notions such as minimax fairness, commonly known as Rawlsian max-min fairness from an utility maximization perspective (Rawls (2001; 2009)), combined with Pareto efficiency, naturally address this no-harm concern (Martinez et al. (2020)).

Recent works study fairness in ML when no information about the protected demographics is available, for example, due to privacy or legal regulations (Kallus et al. (2019)). This is an important research direction and has been identified as a major industry concern (Veale & Binns (2017); Holstein et al. (2019)), since many applications and datasets in ML currently lack demographic records. We therefore study the problem of building minimax Pareto fair algorithms beyond demographics, meaning that not only we lack group membership records but also have no prior knowledge about the demographics to be considered (e.g., any subset of the population can be a valid protected group). This has the advantage of making the model robust to any potential demographic even if they are

unknown at the time of design, or change through time; it is also efficient, since the model offers the best level of service to all the remaining (i.e., non-critical) population.

**Main Contributions.** We analyze *subgroup robustness*, where a model is minimax fair w.r.t. any group of sufficient size (regardless of any preconceived notion of protected groups); we also adhere to the notion of *no-harm* fairness by requiring our minimax model to be Pareto efficient (Mas-Colell et al. (1995)) by providing the best level of service to the remaining population. A model with these characteristics has a performance guarantee even for unidentified protected classes.

We show that being *subgroup robust* w.r.t. an unknown number of groups, where no individual group is smaller than a certain size, is mathematically equivalent in terms of worst group performance to solving a simplified two-group problem, where the population is divided into high and low risk groups, thereby providing a clear means to design "universal" minimax fair ML models. We further show the critical role of the minimum group size by proving that, for standard classification losses (cross-entropy and Brier score), there is a limit to the smallest group size we can consider before the solution degenerates to a trivial, uniform classifier. We additionally study the cost of subgroup robustness when compared to learning a model that is minimax fair w.r.t. predefined demographics.

We then propose *Blind Pareto Fairness* (BPF), a simple training procedure that leverages recent methods in no-regret dynamics Chen et al. (2017) to solve *subgroup robustness* subject to a user-defined minimum subgroup size. Our method is provably convergent and can be used on classification and regression tasks. We experimentally evaluate our method on a variety of standard ML datasets and show that it effectively reduces worst-case risk and outperforms previous works in the area. Although our work is motivated by fairness, subgroup robustness has applications beyond this important problem, see for example (Sohoni et al. (2020a); Duchi et al. (2020)).

## 2    RELATED WORK

A body of work has addressed fairness without (explicit) demographics by using proxy variables to discover the (unobserved) protected population labels (Elliott et al. (2008); Gupta et al. (2018); Zhang (2018)). These methods contrast with our assumptions by relying on a preconceived notion on what the protected demographics are (i.e., the protected demographics are known, but unobserved), since prior knowledge is needed to design useful proxy variables. Moreover, it has been reported that these approaches can exacerbate disparities by introducing undesired bias (Chen et al. (2019); Kallus et al. (2019)); aiming to be fair by inferring protected attributes may be in conflict with privacy or anonymity concerns. These works might need re-training if new protected classes are identified, since a model trained under these conditions may be considerably harmful on an unknown population. This phenomena further supports the value of blind subgroup robustness.

Individual fairness (Dwork et al. (2012)) provides guarantees beyond protected attributes, but requires predefined similarity functions which may be hard or infeasible to design for real-world tasks. The works of (Hébert-Johnson et al. (2017); Kearns et al. (2018)) address fairness w.r.t. subgroups based solely on input features, and while these works greatly extend the scope of the protected demographics, they still rely on labeled protected features for guidance. The work of Sohoni et al. (2020b) partitions the input space to address robust accuracy. We note that partitions,[1] based only on the input space of the model do not modify the solution of risk-based Pareto optimal models, since the optimal classifier for any input value remains unchanged (i.e., there is no conflict between objectives for any value of the input space, see Theorem 4.1 in Martinez et al. (2020)). In our work we consider subgroups based not only on all input features but also on outcome, which broadens the scope to all conceivable subgroups based on the information available to the model. For many risk-based measures of utility, such as crossentropy, Brier score, or L2 regression loss, the optimal classifier can be expressed as a function of the conditional output probability $p(Y|X)$, $X$ being the input (features) and $Y$ the output. In particular, if we only consider groups that introduce covariate shift (i.e., $p(X|A)$ varies across different values of the group membership $A$ but do not change the conditional target distribution ($p(Y|X,A) = p(Y|X)$ for all $A$), then the set of Pareto classifiers only contain one element and the Pareto curve degenerates to the utopia point. By specifically taking outcomes into account in our partition function, we allow for robustness to perturbations on the conditional distribution $p(Y|X,A)$.

---

[1]In this work we consider "partition" and "subgroup" interchangeable.

There are two recent approaches that are the closest to our objective (that of protecting unknown and unobserved demographics). One is distributional robust optimization (DRO) (Hashimoto et al. (2018); Duchi et al. (2020)), where the goal is to achieve minimax fairness for unknown populations of sufficient size. Similar to our work, they minimize the risk of the worst-case group for the worst-case group partition, they use results from distributional robustness that focus the attention of the model exclusively on the high-risk samples (i.e., their model reduces the tail of the risk distribution). However, they do not explicitly account for Pareto efficiency, meaning that their solution may be sub-optimal on the population segment that lies below their high-risk threshold (doing unnecessary harm). The other recent method that tackles the minimax objective is adversarially reweighted learning (ARL) (Lahoti et al. (2020)), where the model is trained to reduce a positive linear combination of the sample errors, these weighting coefficients are proposed by an adversary (implemented as a neural network), with the goal of maximizing the weighted empirical error. This method, while computationally attractive, provides no guarantees about the optimality of the adversary, tradeoffs are also indirectly controlled by adversary capacity and training parameters, which may be less interpretable than constrains on group size or risk thresholds. Our results on critical group size indicate that a sufficiently large capacity adversary should produce a trivial, uniform classifier if no additional constraint is provided. We experimentally compare with these two methods when appropriate.

## 3 PROBLEM FORMULATION

### 3.1 MINIMAX FAIRNESS

We first consider the supervised group fairness classification scenario (Barocas et al. (2019)), where we have access to a dataset $\mathcal{D} = \{(x_i, y_i, a_i)\}_{i=1}^n \sim p(X, Y, A)^{\otimes n}$ containing $n$ i.i.d. triplets. Here $X \in \mathcal{X}$ denotes the input features, $Y \in \mathcal{Y}$ the categorical target variable, and $A \in \mathcal{A}$ group membership. We consider a classifier $h \in \mathcal{H}$ belonging to an hypothesis class $\mathcal{H}$ whose goal is to predict $Y$ from $X$, $h : \mathcal{X} \to \Delta^{|\mathcal{Y}|-1}$; note that $h(X)$ can take any value in the simplex and is readily interpretable as a distribution over labels $Y$. Given a loss function $\ell : \Delta^{|\mathcal{Y}|-1} \times \Delta^{|\mathcal{Y}|-1} \to \mathbb{R}^+$, fairness is considered in the context of a Multi-Objective Optimization Problem (MOOP), where the objective is to learn a classifier that minimizes the conditional group risks $\{R_a(h)\}_{a \in \mathcal{A}}$,

$$\min_{h \in \mathcal{H}}(R_1(h), ..., R_{|\mathcal{A}|}(h)),$$
$$R_a(h) = E_{X,Y|A=a}[\ell(h(X), Y)]. \tag{1}$$

The solution to this MOOP may not be unique (e.g., the optimal classifier of different groups differs), and therefore there is a set of optimal (Pareto) solutions that can be achieved. It is possible that none of these Pareto solutions satisfy some group fairness criteria (e.g., equality of risk), meaning that achieving perfect fairness comes at the cost of optimality (Kaplow & Shavell (1999); Bertsimas et al. (2011)). In this work we do not to compromise optimality, meaning that we do not degrade the performance of a low-risk group if it does not directly benefit another, and consider a minimax fairness approach (Rawls (2001; 2009)), where the goal is to find a Pareto-optimal classifier that minimizes the worst-case group risk,

$$\min_{h \in \mathcal{P}_{\mathcal{A},\mathcal{H}}} \max_{a \in \mathcal{A}} R_a(h), \quad \mathcal{P}_{\mathcal{A},\mathcal{H}} = \{h \in \mathcal{H} : \forall h' \in \mathcal{H} \setminus \{h\} \; \exists a \in \mathcal{A} \text{ with } R_a(h) < R_a(h')\}. \tag{2}$$

$\mathcal{P}_{\mathcal{A},\mathcal{H}}$ represent the set of optimal (Pareto) classifiers in $\mathcal{H}$ given a group set $\mathcal{A}$, meaning that there is no other model in the hypothesis class whose associated group risks are uniformly better for all groups. If the loss function is convex w.r.t. the model $h$, and the hypothesis class $\mathcal{H}$ is a convex set,[2] a linear weighting problem on the conditional group risks ($\min_{h \in \mathcal{H}} \sum \mu_a R_a(h); \sum \mu_a = 1, \mu_a > 0$) characterizes all of the Pareto solutions. Solving Problem 2 is equivalent to finding the weighting coefficients such that a classifier with the minimum worst-case group risk is the solution (Geoffrion (1968); Chen et al. (2017); Martinez et al. (2020)).

### 3.2 BLIND PARETO FAIRNESS

In this work we consider a more challenging problem, namely *Blind Pareto Fairness* (BPF), where the group variable $a$ and the conditional distribution $p(A|X, Y)$ are completely unknown (not just

---

[2]Meaning that for any $h, h' \in \mathcal{H}$ and $\lambda \in [0, 1]$, exists $h_\lambda \in \mathcal{H} : h_\lambda(x) = \lambda h(x) + (1 - \lambda)h'(x) \forall x \in \mathcal{X}$.

unobserved), even at training time. Here the goal is to learn a model that has the best performance on the worst-group risk of the worst partition density $p(A|X,Y)$ ("sensitive" group assignment), subject to a group size constraint ($p(A=a) \geq \rho, \forall a$). We formulate the following new problem,

$$R^* = \min_{h \in \mathcal{P}_{\mathcal{A},\mathcal{H}}} \max_{\substack{p(A|X,Y) \\ s.t.\ p(A=a) \geq \rho,\ \forall a \in \mathcal{A}}} \max_{a \in \mathcal{A}} R_a(h). \tag{3}$$

Here $R^*$ is the minimum worst group error achieved for the worst partition density with known number of partitions $|\mathcal{A}|$. The partition size constraint plays a key role on the solution to Problem 3, as we will show below,[3]. Problem 3 is undetermined in the sense that it admits several worst partition densities and classifiers for $|\mathcal{A}| > 2$. Fortunately, it is possible to show that the minimum worst group error $R^*$ in Problem 3 is the same as the one achieved if we were to consider an alternative formulation where a variable $A \in \{0,1\}$ represents the worst-group risk membership, this is shown in Lemma 3.1. This makes the study of the binary problem attractive when we wish to minimize the number of assumptions we make on the protected groups. Here the objective becomes

$$h^*, p^*(A|X,Y), R^* = \{\arg\} \min_{h \in \mathcal{P}_{\mathcal{A},\mathcal{H}}} \max_{\substack{p(A|X,Y) \\ s.t.\ p(A=a) \geq \rho,\ \forall a \in \{0,1\}}} \max_{a \in \{0,1\}} R_a(h), \tag{4}$$

where we overload the notation $\mathcal{P}_{\mathcal{A},\mathcal{H}}$ in the context of Problem 4 to refer to the Pareto set for a binary group distribution. Lemma 3.1 shows that the minimum worst risk $R^*$ is the same for problems 3 and 4, hence, we focus our analysis on the latter throughout the text. There are two main advantages of working with the binary problem, the first is that finding the worst partition $p(A \mid X,Y)$ for a given $h$ is straightforward when $|\mathcal{A}| = 2$. The second is that in general we may not know the number of groups we wish to be fair to, and this equivalence shows that it is sufficient to specify the minimum size a group must have before it is considered for the purposes of minimax fairness.

**Lemma 3.1.** *Given an hypothesis class $\mathcal{H}$ and a finite alphabet, $\mathcal{A} : |\mathcal{A}| \geq 2$, problems 3 and 4 have the same minimum worst-group risk solution $R^*$ if $\rho \leq \frac{1}{|\mathcal{A}|}$.*

A question that arises from Problem 4 is how the optimal classifier and partition function depend on the partition size. In Lemma 3.2, we show the existence of a critical size $\rho^*$ for standard classification losses (cross-entropy and Brier score) whereby solving Problem 4 for partitions smaller than $\rho^*$ leads to a uniformly random classifier. This result shows that attempting to be minimax fair w.r.t. arbitrarily small group sizes yields a trivial classifier with limited practical utility.

**Lemma 3.2.** *Given Problem 4 with $p(Y|X) > 0\ \forall X,Y$,[4] and let the classification loss be cross-entropy or Brier score. Let*

$$\bar{h}(X) : \bar{h}_i(X) = \frac{1}{|\mathcal{Y}|} \forall X, \forall i \in \{0, ..., |\mathcal{Y}| - 1\},$$

*be the uniform classifier, and let $\bar{h} \in \mathcal{H}$. There exists a critical partition size*

$$\rho^* = |\mathcal{Y}| E_X[\min_y p(y \mid X)] \leq 1$$

*such that solutions to Problem 4, $\forall \rho \leq \rho^*$, are $h^* = \bar{h}$ and $R^* = \bar{R} = \begin{cases} \log |\mathcal{Y}| & if\ \ell = \ell_{CE} \\ \frac{|\mathcal{Y}|-1}{|\mathcal{Y}|} & if\ \ell = \ell_{BS} \end{cases}$.*

*That is, any partitions smaller than $\rho^*$ yield the uniform classifier with constant risk $\bar{R}$.*

It is straightforward to prove that $R^*$ is non-increasing with $\rho$ (see Supplementary Material A.1). A natural question that arises is what is the additional cost in optimality we pay if we apply subgroup robustness instead of optimizing for a known partition. Lemma 3.3 provides an upper bound for the cost of blind fairness, showing that it is at most the difference between $R^*$ and the risk of the baseline model, and can be zero (no cost) if the known group happens to be the worst case partition for the dataset. Moreover, the upper bound decreases with larger group size, and is in no scenario larger than the difference between the risk of the uniform classifier and the baseline classifier for BE and CE losses. Figure 1 shows these concepts graphically.

---

[3]Attempting to be minimax optimal w.r.t. $\rho$ smaller than some value will result in a random classifier, the minimax risk of the partition is directly dependent on $\rho$.

[4]This restriction can be lifted and a similar result holds, see Supplementary Material for details.

**Lemma 3.3.** *Given a distribution $p(X, Y)$ and any predefined partition group $p(A'|X, Y)$ with $A' \in \mathcal{A}'$, $|\mathcal{A}'|$ finite. Let $\hat{h}, \hat{R} = \{\arg\} \min\limits_{h \in \mathcal{H}} \max\limits_{a' \in \mathcal{A}'} R_{a'}(h)$ be the minimax fair solution for this partition and its corresponding minimax risk. Let $h^*$ and $R^*$ be the classifier and risks that solve Problem 4 with $\rho = \min_{a'} p(a')$. Then the price of minimax fairness can be upper bounded by*

$$\max_{a' \in \mathcal{A}'} R_{a'}(h^*) - \hat{R} \leq R^* - \min_{h \in \mathcal{H}} R(h).[5] \tag{5}$$

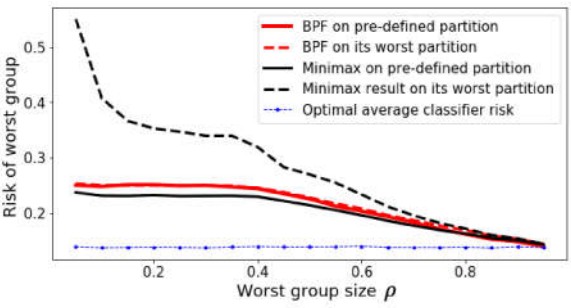 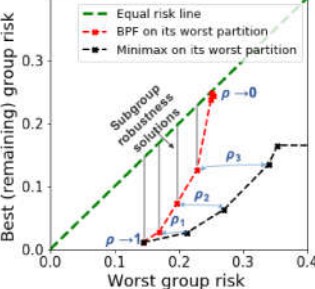

Figure 1: Left figure illustrates Lemma 3.3 by comparing worst case risk with known and unknown partitions. BPF is compared against minimax fairness on their own worst partition and on the predefined partition, optimal average risk ($\min_{h \in \mathcal{H}} R(h)$) is shown in blue. The price of (blind) fairness is the difference in risk we incur when the groups are not known beforehand (BPF vs minimax fairness on predefined partition), represented by the difference between continuous red and black curves. Conversely, optimizing for a known partition (continuous black) leaves another group of the same size with significantly larger risks (dashed black) than what could have been achieved with BPF (dashed red). Right figure highlights the importance of the Pareto optimality constraint, since the space of solutions for subgroup robustness without it contains sub-optimal models that do unnecessary harm on the remaining population. The group risks over the worst partition of the minimax classifier are shown in black for reference; values corresponding to the same group size are linked by arrows. Values shown are Brier scores obtained on a synthetic example, details in Supplementary Material A.2.

In the following section, we provide a practical algorithm that asymptotically,[6] solves Problem 4 and yields a classifier that both minimizes the worst-group risk and is also Pareto-efficient w.r.t. the remaining population. (All proofs are presented in the Supplementary Material A.1.)

## 4 OPTIMIZATION

In order to develop our optimization approach, we begin by showing that for group sizes $\rho \leq \frac{1}{2}$, we can drop the innermost max operator in Problem 4, and we only need to consider the risk for the $a = 1$ partition on a partition of size exactly equal to $\rho$ (i.e., $p(A = 1) = \rho$). This is presented in the following lemma, proved in the Supplementary Material.

**Lemma 4.1.** *Given Problem 4 with minimum group size $\rho \leq \frac{1}{2}$, the following problems are equivalent:*

$$\min_{h \in \mathcal{P}_{\mathcal{A},\mathcal{H}}} \quad \max_{\substack{p(A|X,Y) \\ s.t.\ p(A=a) \geq \rho,\ \forall a \in \{0,1\}}} \max_{a \in \{0,1\}} R_a(h) = \min_{h \in \mathcal{P}_{\mathcal{A},\mathcal{H}}} \quad \max_{\substack{p(A|X,Y) \\ s.t.\ p(A=1) = \rho}} R_{a=1}(h). \tag{6}$$

We note that the right hand side in Eq. 6 is in itself a valuable optimization problem for $\rho > 1/2$, since it equates to efficiently minimizing the risk of the at-risk majority. The Pareto optimality constraint in this scenario is easy to enforce if the base loss function $\ell(h(x), y)$ is both bounded (i.e., $\ell(h(x), y) \leq C \ \forall x, y, h \in \mathcal{X} \times \mathcal{Y} \times \mathcal{H}$), and strictly convex w.r.t. model $h$. The result is shown in the following lemma.

**Lemma 4.2.** *Given the problem on the right hand side of Eq. 6, a convex hypothesis class $\mathcal{H}$, and a bounded loss function $0 \leq \ell(h(x), y) \leq C \ \forall x, y, h \in \mathcal{X} \times \mathcal{Y} \times \mathcal{H}$ that is strictly convex w.r.t its*

---

[5] $R(h) = E_{X,Y}[\ell(h(X), Y)]$.

[6] The algorithm is iterative, we prove convergence to the optimal solution with the number of iterations.

*first input $h(x)$, the following problems are equivalent:*

$$\{\arg\} \min_{\substack{h \in \mathcal{P}_{A,\mathcal{H}}}} \max_{\substack{p(A|X,Y) \\ s.t.\ p(A=1)=\rho}} R_{\alpha=1}(h) \quad = \{\arg\} \min_{h \in \mathcal{H}} \sup_{\substack{p(A|X,Y) \\ s.t.\ p(A=1)=\rho \\ p(A=1|X,Y)>0,\ \forall X,Y}} R_{\alpha=1}(h). \tag{7}$$

Where we explicitly add $\{\arg\}$ to both sides of the equivalence to indicate that these problems are also equivalent in terms of the models $h$ that achieve these minimax solutions. The BS loss satisfies both conditions in Lemma 4.2, CE loss also satisfies these conditions if we restrict the hypothesis set such that $h \in \mathcal{H} \subseteq \{h : h_i(x) \geq \zeta > 0 \forall i \in [\mathcal{Y}], x \in \mathcal{X}\}$ (i.e., the classifier assigns a minimum label probability for all values). the $\ell_2$ loss over a bounded set also satisfies these conditions. To deal with the supremum constraint on the distribution in practice, we slightly limit adversary capacity and ensure $p(A = 1|X, Y) \geq \epsilon > 0 \forall X, Y \in \mathcal{X} \times \mathcal{Y}$.

We solve Problem 7 using no-regret dynamics (Freund & Schapire (1999)), the solution is the Nash equilibrium of a two-player zero-sum game, where one player, the adversary, iteratively proposes partition distributions $p(A|X, Y)$, the modeler then responds near optimally with a model $h$, and incurs loss $R_1(h)$. Based on the history of losses, the adversary iteratively refines its proposed partition function into the worst-case partition.

To solve the above problem with parameter $\epsilon > 0$, we leverage the theoretical results presented in Chen et al. (2017) for improper robust optimization of infinite loss sets with oracles. We first present the results in terms of a finite dataset with $n$ samples; assume that both players have access to $\{x_i, y_i\}_{i=1}^n \sim P(X, Y)^{\otimes n}$, and let $t \in \{0, \ldots, T\}$ indicate the current round of the zero-sum game. In each round $t$, the modeler produces a classifier $h^t$ and the adversary proposes an empirical distribution of $p(A|X, Y)$, denoted as $\alpha^t = \{\alpha_i^t\}_{i=1}^n : \alpha_i^t \in [\epsilon, 1], \sum_i \frac{\alpha_i^t}{n} = \rho$, where $\rho$ is the minimum partition size. The empirical risk (cost) of round $t$ is $L^t = L(h^t, \alpha^t)$, with

$$L(h, \alpha) \quad = \frac{\sum_{i=1}^n \alpha_i \ell(h(x_i), y_i)}{n\rho}. \tag{8}$$

In order to find the Nash equilibrium of this game, we use projected gradient descent on the adversary, while the modeler uses approximate best response with a Bayesian oracle $h^t = M(\alpha^t)$. In particular, we use a variant proposed in Chen et al. (2017) for robust non-convex optimization. Algorithm 1 shows the proposed approach.

---

**Algorithm 1** Blind Pareto Fairness

---

**Require:** Inputs: Dataset $\{(x_i, y_i)\}_{i=1}^n$, minimum partition size $\rho$

**Require:** Hyper-parameters: Number of rounds $T$, parameter $\eta$, adversary boundary coefficient $\epsilon > 0$, $\gamma$-approximate Bayesian solver $M(\cdot) \simeq \arg\min_{h \in \mathcal{H}} L(h, \cdot)$

Initialize $\alpha^0 = \hat{\alpha} = \{\rho\}_{i=1}^n$

Initialize classifier and loss

$h^0 = M(\hat{\alpha}), L^0 = L(h^0, \hat{\alpha})$

**for** round $t = 1, \ldots, T$ **do**

    *Adversary updates its partition function by gradient descent and projection:*

    $\alpha^t \leftarrow \alpha^{t-1} + \eta \nabla_\alpha L(h^t, \hat{\alpha}) = \alpha^{t-1} + \eta \frac{\ell(h^t, y)}{n\rho}$

    $\hat{\alpha} \leftarrow \prod_{\alpha:\alpha_i \in [\epsilon,1], \sum_i \frac{\alpha_i}{n}=\rho} (\alpha^t)$

    *Solver provides a model that approximately solves for the current partition:*

    $h^t \leftarrow M(\hat{\alpha})$

**end for**

**return** No-regret classifier $h^T$

---

The proposed Algorithm 1 in an instantiation of Algorithm 3 in (Chen et al. (2017)) for oracle efficient improper robust optimization with infinite loss sets. To implement the projection operator $\prod_{\alpha:\alpha_i \in [\epsilon,1], \sum_i \frac{\alpha_i}{n}=\rho} (\cdot)$, we use Dykstra's projection algorithm (Boyle & Dykstra (1986)), using the fact

that the set can be seen as the intersection of the $[\epsilon, 1]$ hypercube and the $\langle \alpha, \mathbf{1} \rangle = n\rho$ plane, where both of these sets have known and exact projection functions individually. We can then immediately leverage their results (Theorem 7 in Chen et al. (2017)) to show that the algorithm converges (proof presented in the Supplementary Material).

**Lemma 4.3.** *Consider the setting of Algorithm 1, with parameter $\epsilon > 0$ and $\eta = \max_{\alpha \in \{\alpha : \alpha_i \in [\epsilon, 1], \sum_i \frac{\alpha_i}{n} = \rho\}} \frac{||\alpha||_2}{\sqrt{2T}} \leq \sqrt{\frac{n\rho}{2T}}$, and $L$ a 1-Lipschitz function w.r.t. $\alpha$, let $P$ be a uniform distribution over the set of models $\{h^1, \ldots, h^T\}$, and let $R^*$ be the minimax solution to the loss presented in Eq. 8. Then we have*

$$\max_{\alpha : \alpha_i \in [\epsilon, 1], \sum_i \frac{\alpha_i}{n} = \rho} \mathbb{E}_{h \sim P} L(h, \alpha) \leq \gamma R^* + \sqrt{\frac{2n\rho}{T}}.$$

As in Chen et al. (2017), we use $h^t$ instead of the ensemble $\{h^1, \ldots, h^T\}$. We use stochastic gradient descent (SGD) as our $\gamma$-approximate Bayesian oracle, We note that the 1-Lipschitz constraint can be relaxed to any $G$-Lipschitz function by working through the no regrets guarantees for projected gradient descent of $G$-Lipschitz functions in the proof provided in Chen et al. (2017).

Figure 2 shows how group risks evolve across game rounds, results are shown for several minimum partition size values. We observe that cross-entropy (CE) on the worst group gradually converges to a value that depends on $\rho$, the rounds trade performance between low and high risk groups.

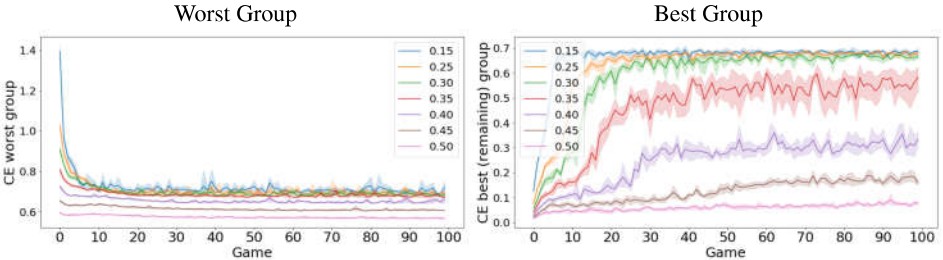

Figure 2: Risk trade-off between low and high group risks in the BPF algorithm over the UCI adult Dua & Graff (2017) dataset as a function of game rounds. Left figure indicates worst group cross-entropy, right corresponds to low risk groups. Traces are shown for several $\rho$ values. Small $\rho$ values make the risk of the worst group converge to that of the random classifier, as expected from Lemma 3.2, larger values show a more stratified behaviour. We observe how performance between worst and best groups is traded off between game rounds.

## 5 EXPERIMENTAL RESULTS

We experimentally validate our methods and theoretical results on a variety of standard datasets, we compare performance against DRO (Hashimoto et al. (2018)), ARL (Lahoti et al. (2020)), and a baseline (naive) classifier. We show the trade-offs of each method on their worst group and the remaining population. The baseline method is, as expected, the one that performs best on the low-risk population, but it suffers from large, fat tails in terms of loss distribution. We also show how both DRO and BPF empirically achieve the theoretical results laid in Lemma 3.2, with BPF having better results on the low-risk population than DRO, owing to the Pareto optimality constraint.

**Datasets.** We used four standard fairness datasets for comparison. The UCI Adult dataset (Dua & Graff (2017)) which contains $48,000$ records of individual's annual income as well as 13 other attributes, including race, gender, relationship status, and education level. The target task is income prediction (binary, indicating above or below $50K$). The Law School dataset (Wightman (1998)) contains law school admission data used to predict successful bar exam candidates based on various factors including family income, race, and gender. The COMPAS dataset (Barenstein (2019)) which contains the criminal history, serving time, and demographic information such as sex, age, and race of convicted criminals. The goal is prediction of recidivism per individual.[7] Lastly we used the

---

[7]This dataset is the source of extensive and very legitimate controversy in the fairness community, and is here used for benchmarking only.

MIMIC-III dataset, which consists of clinical records collected from adult ICU patients at the Beth Israel Deaconess Medical Center Johnson et al. (2016). The objective is predicting patient mortality from clinical notes. We analyze clinical notes acquired during the first 48 hours of ICU admission following the pre-processing methodology in Chen et al. (2018), ICU stays under 48 hours and discharge notes are excluded from the analysis. Tf-idf statistics on the $10,000$ most frequent words in clinical notes are taken as input features.

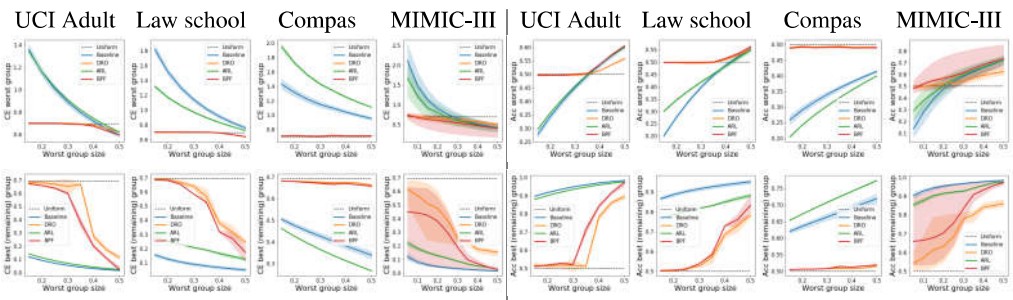

Figure 3: Cross-entropy (CE) and accuracy (Acc) metrics on best and worst groups as a function of group size for BPF, DRO, ARL, and baseline classifiers, random classifier shown for reference. Results are provided for UCI adult, law school, COMPAS, and MIMIC-III datasets. Cross-entropy of both ARL and baseline classifiers for the worst group are very large for small group size, DRO and BPF both approximate the theoretical result shown in Lemma 3.2. The main difference between DRO and the proposed BPF is that BPF exhibits better results on the best group partition than DRO for the same level of worst group performance, owing to the Pareto restriction on the BPF classifier resulting in no-unnecessary-harm for any group (see also right panel of Figure 1). Accuracy results largely mimic the observations on the cross-entropy metric.

**Setup and Results.** We train BPF for 8 minimum group sizes $\rho = \{0.15, 0.20, \ldots, 0.5\}$, we report cross-entropy loss and accuracy,[8] on the worst partition of the dataset (i.e., average over the worst $100 * \rho\%$ samples based on cross-entropy loss), the values for the remaining low risk group is also reported (to evaluate optimality). DRO models were trained on 20 equispaced values of their threshold parameter ($\eta \in [0, 1]$). For ARL, we use the code provided in (Lahoti et al. (2020)); since it lacks a tuning hyper-parameter, besides the original setup, we tried four configurations for their adversarial network (adversary with 1 to 4 hidden layers of 512 units each). The classifier architecture for BPF, ARL, and DRO was standardized to a 4-layer MLP with 512 hidden units. In all cases we use cross-entropy loss and same input data. Results correspond to the best hyper-parameter for each group size; mean and standard deviations are computed using 5-fold cross-validation.

Figure 3 shows the performance of the best and worst groups across partition sizes. Both DRO and BPF recover results close to the random classifier for the smaller group sizes, which aligns with the results shown in Lemma 3.2, that is, below a certain partition size (e.g., $\rho \simeq 0.3$ for adult dataset) the average cross-entropy of the worst group is the risk of the uniform classifier ($\log 2$). We observe that, in general, BPF obtains better results for the low risk group than DRO for the same worst group risk. ARL slightly reduces the worst group risk compared to the baseline classifier, but is in general not able to reduce worst-case risk significantly.

Although none of the compared models address disparities along predefined populations, in some cases they are nonetheless able to improve the worst group performance on the set. Consider the Adult dataset, an unbalanced problem ($\approx 23.6\%$ of the samples present an income higher than $50k$) where sample accuracy may be a misleading statistic (e.g., baseline classifier achieves $> 81\%$ accuracy across male and female populations). Table 1 shows accuracy conditioned on both gender and income, and on race and income for each competing method. We observe that both BPF and DRO achieve results close to the uniform classifier for small partition sizes as expected, and no predefined group larger than their partition size gets a worse-than-random performance. In many cases, the results for BPF is better than DRO for each protected attribute (at same $\rho$ value). We also observe that on several minorities, the BPF model provides the best utility values out of all the competing methods, BPF is also the best model at preserving worst group performance.

---

[8]Accuracy is computed on the randomized classifier $\hat{Y} \sim h(X)$.

Table 2 shows how target labels and predefined sensitive groups are represented in the high risk group identified by BPF. We observe that, for low partition sizes, outcomes are balanced across groups (in concordance with Lemma 3.2). As the partition size increases, the composition of the high risk group becomes more similar to the base distribution. Similar results to tables 1 and 2 are provided in Supplementary Material A.3 for law school, COMPAS and MIMIC-III datasets.

| Group | Prop (%) | baseline | ARL | DRO .15 | BPF .15 | DRO .4 | BPF .4 | DRO .5 | BPF .5 |
|---|---|---|---|---|---|---|---|---|---|
| Race/Income (1 if $\geq$50k, 0 if $<$50k ) | | | | | | | | | |
| Male/0 | 46.7 | 82.7±0.1 | 81.3±0.3 | 51.2±0.9 | 50.7±1.3 | 68.4±1.6 | 70.2±2.1 | 74.0±1.6 | 78.9±0.4 |
| Male/1 | 20.0 | 59.3±0.9 | 59.4±0.5 | 50.6±0.3 | 51.1±1.2 | 54.2±0.8 | 57.5±1.3 | 55.3±1.2 | 58.8±0.1 |
| Female/0 | 29.7 | 94.0±0.2 | 93.5±0.2 | 52.2±1.7 | 50.8±2.1 | 76.8±1.8 | 73.7±1.7 | 84.9±1.5 | 90.2±0.6 |
| Female/1 | 3.6 | 50.8±2.2 | 50.0±0.5 | 50.6±0.3 | 51.6±2.3 | 50.5±1.1 | 52.5±0.9 | 50.1±1.4 | 52.2±1.0 |
| Ethnicity/Income (1 if $\geq$50k, 0 if $<$50k ) | | | | | | | | | |
| White/0 | 64.2 | 86.2±0.2 | 85.2±0.2 | 51.4±1.1 | 52.7±2.1 | 70.7±1.7 | 68.3±2.4 | 77.1±1.5 | 82.5±0.5 |
| White/1 | 21.4 | 58.6±1.1 | 58.4±0.4 | 50.6±0.3 | 51.1±2.3 | 53.9±0.8 | 57.0±1.3 | 54.9±1.2 | 58.1±0.1 |
| Black/0 | 8.5 | 92.5±0.5 | 91.9±0.4 | 52.8±1.9 | 51.0±2.6 | 78.2±1.7 | 72.1±2.5 | 85.6±1.9 | 89.1±0.7 |
| Black/1 | 1.1 | 52.2±0.0 | 51.9±1.0 | 50.6±0.3 | 50.8±2.3 | 51.4±1.9 | 53.2±1.4 | 51.7±2.4 | 54.0±0.7 |
| Asian-PacI/0 | 2.1 | 86.5±0.7 | 83.8±1.1 | 51.0±0.7 | 50.6±3.1 | 71.3±2.8 | 68.1±2.1 | 77.5±3.2 | 80.9±0.7 |
| Asian-PacI/1 | 0.8 | 59.3±1.5 | 60.2±2.0 | 50.8±0.4 | 52.3±2.2 | 52.8±2.6 | 62.2±2.1 | 53.2±3.0 | 58.9±1.9 |
| Other/0 | 1.5 | 93.3±0.2 | 93.2±0.4 | 51.7±1.3 | 50.8±2.1 | 77.6±1.9 | 73.9±2.0 | 85.5±2.2 | 89.5±1.5 |
| Other/1 | 0.3 | 41.4±2.1 | 39.4±2.7 | 50.5±0.3 | 51.4±2.5 | 44.9±2.7 | 46.7±2.1 | 42.5±3.4 | 44.6±3.1 |

Table 1: Accuracy across gender and ethnicity partitions (groups given no special consideration by the algorithms) in the Adult dataset for ARL, DRO and BPF models for varying partition sizes.

| Group | Prop(%) | BPF .15 | BPF .3 | BPF .4 | BPF .5 |
|---|---|---|---|---|---|
| Proportion on Worst Partition, Ethnicity/Income | | | | | |
| White/0 | 64.2 | 41.8±0.9 | 44.8±0.4 | 48.9±0.1 | 53.5±0.0 |
| White/1 | 21.4 | 47.3±1.0 | 44.9±0.6 | 41.0±0.0 | 35.6±0.1 |
| Black/0 | 8.5 | 2.7±0.2 | 3.0±0.0 | 3.4±0.0 | 4.6±0.0 |
| Black/1 | 1.1 | 2.9±0.1 | 2.5±0.0 | 2.2±0.0 | 1.8±0.0 |
| Asian-PacI/0 | 2.1 | 1.8±0.1 | 1.7±0.2 | 1.8±0.1 | 1.9±0.0 |
| Asian-PacI/1 | 0.8 | 2.0±0.0 | 1.8±0.0 | 1.6±0.0 | 1.4±0.0 |
| Other/0 | 1.5 | 0.4±0.0 | 0.4±0.0 | 0.5±0.0 | 0.8±0.0 |
| Other/1 | 0.3 | 1.0±0.0 | 0.8±0.0 | 0.6±0.0 | 0.5±0.0 |

Table 2: Demographic composition of worst groups as a function of minimum partition size on the Adult dataset. BPF homogenizes outcomes across partitions and protected attributes. For larger group sizes, the demographics of the partition approach that of the baseline population.

## 6 DISCUSSION

In this work we analyze subgroup robustness, particularly in the context of fairness without demographics or labels. Our goal is to recover a model that minimizes the risk of the worst-case partition of the input data subject to a minimum size constraint, while we additionally constrain this model to be Pareto efficient w.r.t. the low-risk population as well. This means that we are optimizing for the worst unknown subgroup without causing unnecessary harm on the rest of the data. We show that it is possible to protect high risk groups without explicit knowledge of their number or structure, only the size of the smallest one, and that there is a minimum partition size under which the random classifier is the only minimax option for cross-entropy and Brier score losses.

We propose BPF, an algorithm that provably converges to the Pareto minimax solution. Our results on a variety of standard fairness datasets show that this approach reduces worst-case risk as expected, and produces better models than competing methods for the low-risk population, thereby avoiding unnecessary harm.

If a policymaker has a desired risk tradeoff instead of a target group size, we can search for the smallest partition size achieving this tradeoff using the proposed BPF; this now guarantees that the recovered model can satisfy this risk tradeoff for the worst possible partition up to size $\rho$, and for any smaller partition size there exists a partition such that this tradeoff is violated.

Future work includes incorporating additional domain-specific constraints on the worst partition and developing an algorithm that combines BPF with knowledge about some subgroups that must be protected as well.

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
