# OpenReview forum: "Blind Pareto Fairness and Subgroup Robustness"
_ICLR.cc/2021/Conference — Reject_

### Official Review · AnonReviewer4 · 2020-10-28
**Interesting problem, but unable to verify correctness of the proposed approach**

**Rating:** 6
**Confidence:** 4

**Review:**

### POST-REVISION
Thanks for revising your algorithm and clarifying its theoretical properties.

I think the changes made to the algorithm do seem substantial: you've changed a complicated combination of multiplicative and best response updates on alpha to a simpler projected gradient update  (I am assuming that the experimental results have also been revised accordingly). The description is now easier to follow and the convergence results now follow directly from Chen et al.

I'm raising my score to 6, but still find the novelty in the formulation to be somewhat limited.

One pending concern is that the algorithm maintains one parameter \alpha_i per training example, which can be prohibitively expensive for large datasets. Of course, one way to alleviate this difficulty would be to replace \alpha_i with a parameterized function of the features, which however, would make your formulation very similar to Lahoti et al. (2020).

Is there a way you could measure the violation in the pareto constraint in your experiments, to showcase how the classifiers learned by your method are different from those learned by the approach of Lahoti et al?
******

The authors seek to learn a classifier which is a pareto-efficient and achieves the min-max risk across different subgroups. They are particularly interested in a setting where neither the group memberships nor the number of groups is known a priori. Prior work by Martinez et al. (2020) propose an approach for learning min-max pareto fair classifier when the group identities are known beforehand. This paper's proposal is to formulate a min-max optimization problem: like Martinez et al. the minimization is over the space of pareto-efficient classifiers, whereas the maximization is over all (soft) partitions of the data into two groups.

The authors show that the solution to this problem degenerates into a trivial classifier when the size of the smallest group is below a certain threshold. They then propose an algorithm inspired from Chen at al. (NeurIPS 2017) to solve the min-max problem.

I like the problem formulation and find the problem setup to be practically relevant (given that there's an increasing emphasis in the fairness community on addressing scenarios where the protected group information is noisy or unknown).

However, I'm unable to verify correctness of algorithm and guarantee
- The authors say they adapt Algorithm 3 from Chen at al., but I find the proposed algorithm to be different from the one in Chen et al. Unfortunately, the textual description doesn't delve into any of these differences. I am also unable to see how the no-regret guarantee the authors provide in Lemma 4.1 follows from results in Chen et al. They provide a one line proof stating "Immediate application of Theorem 11 in Chen et al. (2017)" but I don't see how Theorem 11 directly applies to the specific problem setup the authors consider.

Below, I elaborate the discrepancy I see with Chen et al.

Algorithm 3 in Chen et al., which the authors adapt, consider the following setup: there's an objective L(x, w) which is possibly non-convex in "x" and is convex in "w", and the goal is to minimize L over x and maximize it over w. They assume access to an oracle that can minimize L(., w) over "x" for a fixed "w", and provide an algorithm which performs gradient ascent updates on "w" and performs full optimization over "x" using the oracle. They show convergence guarantees for this algorithm.

The min-max problem considered in the present paper also has two sets of parameters "alpha" and classifiers "h". The objective is non-convex in alpha (see eq. 6), and convex w.r.t. "h" (Sec 3.1). So the natural analogue of the algorithm of Chen et al. for this setup, would be to perform full optimization over alpha (which the authors do), and to perform (projected) gradient updates on "h". However, instead what the authors use for "h" is some form of a multiplicative weights update, which I find confusing. What you have is a multiplicative update again on alpha, and an invocation of another algorithm by Martinez et al.  to learn a pareto-efficient classifier. Was the idea to perform a multiplicative update on "h" and project it back to the space of pareto-efficient classifiers?

I agree that like gradient ascent, multiplicative weights (MW) also enjoys no-regret guarantees but to apply MW, won't you need to maintain an explicit weight for each classifier "h"? Instead, why is the MW update applied again to the alpha's?

At the very least, I think authors need to elaborate on how exactly they apply the algorithm from Chen et al. The current discussion is light on the details.

---

> ### Author Response · Authors · 2020-11-25
> **Response**
>
> We thank the reviewer for the very careful reading and study of the proposed optimization method. We carefully reviewed their comments and agree that the presentation of the optimization algorithm needed more exposition and clarity. While reviewing the framework to clarify this comment, we noticed that the proposed methodology and algorithm could be simplified. We added further analysis/explanations, showcasing that our approach is fully in line with the theoretical results proposed in Chen et al. The method, as well as the expanded explanations and proofs are provided in the Optimization section and Supplementary Material. This includes two new Lemmas presented in the main body of the paper that we think improve the presentation of the theoretical foundation of the proposed optimization.
>
> All experimental results are now shown with the simplified algorithm, with no significant changes in any particular result. We reiterate that, although the optimization method was not originally the main focus of the paper, we are very thankful for the careful revision, and we are confident that the current optimization method and its presentation are cleaner, more clearly in line with the theory provided in Chen, and is now presented in a way that makes the key insights of the mapping of our problem into a provably convergent algorithm significantly more clear.

---

### Official Review · AnonReviewer1 · 2020-10-28
**The authors explore how group fairness can be maintained when the groups are not predefined.**

**Rating:** 6
**Confidence:** 4

**Review:**

## Summary ##

This paper considers the relevant problem of group fairness in ML when there are no predefined groups. They aim to provide an algorithm that outputs a classifier that minimizes the risk of any group (of sufficient size) while at the same time being Pareto efficient. They do this by first showing that this problem is equivalent to the case when there are only two groups and then using no-regret dynamics to provide an algorithm which asymptotically converges to the optimal solution.

They also show empirically, using multiple datasets, that their algorithm is marginally better than the state of the art in terms of fairness and in some cases, efficiency.


## Strengths ##
This is clearly a very relevant problem since (as they mention) in a lot of cases, we may not have the group labels for each point in the dataset. Furthermore, being robust to any subgroup allows for the algorithm to be fair to groups who have not yet been declared as protected groups.

The algorithm asymptotically converges to the optimal solution. This algorithm not only minimizes the risk of the worst case group but also does not do any unnecessary harm to other groups while ensuring fairness.
The experimental section is very comprehensive and compares their algorithm to the state of the art using multiple datasets. It shows that in almost all of them, their algorithm does marginally better than the state of the art. This section also gives us an idea about the trade-off between efficiency and fairness.
Weaknesses:

* The improvement over the current state of the art in this problem (DRO) is very marginal and in some cases (the Compas Dataset), DRO does slightly better than BPF.
* There is no discussion about the result in Lemma 3.3. It is unclear whether this bound is reasonable or whether the loss in efficiency is significant.
* Lastly, in a lot of cases, the accuracy of groups which are not the worst-off group is significantly worse than that of the baseline while the accuracy of the worst off group is only marginally better than that of the baseline. While it is clear that this loss in accuracy is necessary to improve the accuracy of the worst off group, there is very little discussion on why this tradeoff is worth it, or whether there exists another way to address this trade-off.

## Typos ##
* Unnecessary Parenthesis around Utsun et al (2019) (Introduction)
* $Y$ should be $\mathcal Y$ (Section 3.1, Line 6)
* $\Delta^{|Y| - 1}$ should be $\Delta^{|Y|}$ (Section 3.1)
* $p(A = a) \ge \rho$ should be $p(A = 1) \ge \rho$ (Proof of Lemma 3.1, Line 6)
* There is a typo in the equation $P(Y|A = 0, X) = \dots$ (Proof of Lemma 3.2)
* $a^*$ is not defined (Proof of Lemma 3.3)

## Other structural feedback ##
The classes of hypothesis functions that can be used is unclear throughout. It would help to discuss it for the readers who have not read Martinez et al (2020)

## After Rebuttal ##

I thank the authors for their clarifications and efforts to improve their work. I still support acceptance.

---

> ### Author Response · Authors · 2020-11-25
> **Response**
>
> We appreciate the constructive feedback from the reviewer; we address their main concerns below and indicate how they have been implemented in the revised paper.
>
> * While the improvements versus DRO are not large on these datasets, BPF consistently outperforms DRO on the best group, and is equal to DRO on the worst group; this can be seen in Figure 3. BPF and DRO both tackle similar problems, and can indeed be used almost interchangeably to reason about these issues, but the additional Pareto efficiency constraint in BPF (which is not present in DRO) ensures that, for any given  worst group performance, the BPF risk on the best group is lower or equal than the corresponding risk for DRO. To strengthen our comparisons, we included results on a new dataset where the objective is predicting ICU patient mortality from clinical notes  (MIMIC III dataset). see Figure 3.
>
> * The results on Lemma 3.3 essentially depend on the given (fixed) subgroup and the base data distribution. What we can glean from this lemma is that the potential cost of blind fairness increases when the target group size is small. Note that the right hand side of lemma 3.3 is itself upper bounded (for crossentropy and brier score) by the difference in risk between the uniform classifier and the baseline classifier. Meaning that under no scenario would blind fairness cost more than that difference. The efficiency decrease can also be exactly equal to 0 if the known group happens to be the worst case partition for the dataset. This extended discussion has been added into the paper.
>
> * We added comments in the discussion section addressing the loss in accuracy discussed by the reviewer. While accuracy is not the primary risk being minimized, we can state that the crossentropy tradeoffs achieved by BPF are theoretically the best achievable tradeoffs for the dataset. The question on whether this occasionally large tradeoff is desirable is outside the scope of the paper, but we can provide an attractive dual interpretation for these tradeoffs as the following:
>
> Given a desired risk tradeoff, we can find the smallest partition size achieving this tradeoff using BPF, this now guarantees that the recovered model can satisfy this risk tradeoff for the worst possible partition up to size $\rho$; and for any smaller partition size, there exists a partition such that this tradeoff is violated.
>
> This comment is added in the discussion section of the paper.
>
> * We have corrected all the typos save for the simplex notation, since we choose by convention to denote the degrees of freedom of the simplex in the superindex (e.g., a simplex over 3 categorical variables is denoted \Delta^2)

---

### Official Review · AnonReviewer5 · 2020-11-09
**Weak acceptance**

**Rating:** 6
**Confidence:** 4

**Review:**

##########################################################################

Summary:
The authors analyze the space of solutions for worst case fairness beyond demographics, and propose Blind Pareto Fairness (BPF), a method that leverages no-regret dynamics (i.e., Multiplicative Weight Update) to recover a fair minimax classifier that reduces worst-case risk of any potential subgroup of sufficient size (learnt over worst-case partitions, such that the smallest size/density is at least a minimum), and guarantees that the remaining population receives the best possible level of service. They show promising experimental results over three data sets: UCI dataset on Adult Income, Law School dataset, and Compas Dataset.


##########################################################################

Reasons for score:
The idea for establishing performance for unknown groups is interesting, simply because the set of protected attributes (e.g., nationality, race, genetic information) is evolving with time. In this regard, I find the premise of the paper interesting. To obtain a minimax classifier, the authors deploy a multiplicative weight update based method, which is somewhat standard.  Their setting is a bit incremental compared to multiple recent papers on minimax Pareto fairness guarantees, and not motivated beyond unknown groups (see cons). I therefore support (weak) acceptance of this work since they move away from "given groups" and show an improvement in the experiments for the worst-off group.

##########################################################################

Pros:

BPF addresses fairness beyond demographics, that is, it does not rely on predefined
notions of at-risk groups, neither at train nor at test time.

Experimental results show that the proposed framework improves worst-case risk in multiple standard datasets, while simultaneously providing better levels of service for the remaining
population, in comparison to competing methods.


##########################################################################

Cons:

Comparisons from related work are weak. For example, the authors claim that "In our work
we consider subgroups based not only on all input features but also on outcome, which broadens the
scope to all conceivable subgroups based on the information available to the model." in comparison to Martinez et. al, which already considers minimax pareto fairness, for instance. Why are subgroups based on outcomes relevant? Why would it make sense in any application to consider based on outcomes.

In comparison to Duchi et. al 2020, and Hashimoto et a. 2018, the authors say that "can be sub-optimal on the population segment that lies below their high-risk threshold (doing unnecessary harm)", however one can imagine that in some applications these high risk samples are in fact more important than the low risk ones. For which applications do the authors argue that distribution independent results are more useful, compared to these recent distributionally robust results?

##########################################################################

Questions during rebuttal period:


Please address and clarify the cons above


#########################################################################

Typos:

---

> ### Author Response · Authors · 2020-11-25
> **Response**
>
> We appreciate the constructive feedback from the reviewer; we address their main concerns below and indicate how they have been implemented in the revised paper.
>
> * While we build on previous work on Pareto fairness, this paper has a number of major novelties which -in our view- are not incremental, including:
> - Considering unknown subgroups is a relatively new and important direction of research, not just in fairness but in robustness as well.
> - Some of our theoretical results are critical to shed further light onto this problem, e.g., Lemma 3.2 indicates that there is a minimal size beyond which we cannot control fairness/robustness and Lemma 3.3 explicitly bounds the price of blind fairness versus fairness with respect to known groups.
>
> * The main appeal of considering subgroups based on outcomes is the following For many risk-based measures of utility, such as crossentropy, Brier score, or \ell_2 regression loss, the optimal classifier can be expressed as a function of the conditional output probability p(Y|X), X being the input (features) and Y the output. In particular, if we only consider groups that introduce covariate shift (i.e., p(X|A) varies across different values of the group membership A) but do not change the conditional target distribution (p(Y|X,A) = p(Y|X) for all A ), then the set of Pareto classifiers only contain one element and the Pareto curve degenerates to the utopia point. By specifically taking outcomes into account in our partition function, we allow for robustness to perturbations also on the conditional distribution p(Y|X,A). This has now been clarified at the end of the 2nd paragraph of Section 2.
>
> * Regarding comparison with Duchi et al. 2020 and Hashimoto el al. 2018. The Pareto constraint on our classifier can be interpreted in another way. Namely, for any classifier that satisfies a desired performance objective over the high loss samples (the main objective of Minimax Fairness), Pareto optimality allows us to select the classifier that simultaneously has the best performance on low-loss samples. There are only benefits to be had by restricting the search space to Pareto optimal solutions. This can be seen in Figure 3, where BPF obtains the same results over the high-loss groups as DRO, but incurs lower losses over the low-loss samples on the same model. This has been further clarified in the captions of figure 3, as well as other parts of the paper where the no-unnecessary-harm attribute of the Pareto framework is discussed, (e.g., in the discussion).
>
> Since both the Blind Pareto Fairness and DRO methods can be applied to essentially the same set of problems, unless there is no interest in providing a better level of service to the low-loss population, there is no reason to not using BPF over other non-pareto distributionally robust methods.
>
> All of these comments have been interleaved throughout the revised manuscript.

---

### Author Response · Authors · 2020-11-25
**Overall response to AC and reviewers**

We thank all the reviewers for the very constructive feedback and we appreciate the AC for her/his time with our paper; they have helped us to improve the presentation and to further clarify the contributions of the paper. We detail below how we have addressed their comments, but before that we want to reiterate the main contributions of the paper:

•	We analyze the problem of fairness and subgroup robustness when the subgroups are not known in advance, but the size of the smallest group can be bounded.

•	The formulation explicitly addresses and guarantees that no subgroup receives unnecessary harm other than what is strictly necessary to improve the performance of the worst subgroup.

•	We analyze the tradeoff between group fairness with known and unknown subgroups, and bound the price of blind fairness.


•	We leverage recent results from robust optimization of non-convex objectives and provide a convergent optimization algorithm addressing the proposed problem.

We consider that the contributions presented in this work provide valuable insight into some of the fundamental properties of subgroup robustness and blind (unknown demographic) fairness; an area that has recently started to attract interest.


We have included  an additional dataset (predicting ICU patient mortality from clinical notes in MIMIC III dataset) on the experimental section.

Below we provide a detailed address to all the comments

---

### Decision · Program_Chairs · 2021-01-07
**Final Decision**

**Decision:**

Reject

**Comment:**

This paper studies the problem of Pareto fairness without having pre-defined protected groups. The reviewers agree that the problem studied here is interesting and relevant. During the initial review period, reviewers identified a major correctness issue. The authors have then substantially changed the algorithm and experiments in the rebuttal period in order to address the issues. Now the convergence result in the paper follows more directly from the prior work of Chen et al. Overall, the technical novelty of the paper appears to be limited. Even though the authors have also strengthened the related work discussion, they should also consider discussing the comparison between their work with that of Lahoti et al., as suggested by one of the reviewers.